# ADAPTIVE MASKING ENHANCES VISUAL GROUNDING

## ABSTRACT

Humans excel at recognizing objects under incomplete information by focusing on their most salient features. Inspired by this capability, we present **IMAGE** (**I**nterpretative **MA**sking with **G**aussian Radiation Mod**E**ling), a novel training paradigm for visual grounding that selectively obscures salient regions. By compelling models to infer objects from suboptimal cues, IMAGE mimics human adaptability in scenarios where critical features are absent. We propose a progressive training strategy that gradually increases the masking ratio, compelling the model to extract essential object attributes rather than memorizing all possible features. Experiments on standard visual grounding benchmarks demonstrate notable improvements in zero-shot and low-shot scenarios, with IMAGE seamlessly integrating into existing architectures. The method's training-only operation ensures zero added computational cost during deployment, offering a practical pathway toward robust, data-efficient visual grounding.

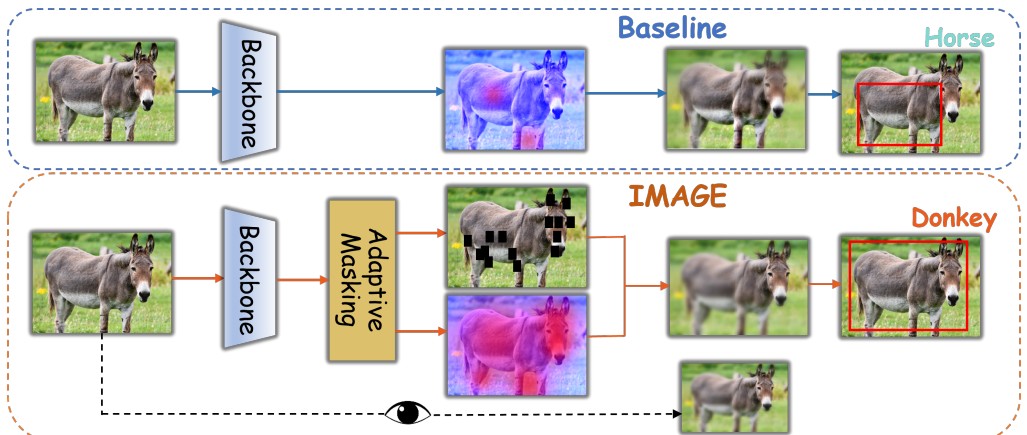

Figure 1: Our IMAGE method is inspired by human perception; by masking key details of objects, we encourage the model to learn more robust representations.

## 1 INTRODUCTION

*"To see the world in a grain of sand,"*
        – William Blake, *Auguries of Innocence*

In human perception, key details of an object guide our understanding even when certain features are missing or occluded. Inspired by this, recent research in visual grounding has emphasized low-shot learning, where models must identify novel objects with minimal labeled data. This paradigm is crucial in settings like autonomous driving Rezaei & Shahidi (2020) and embodied AI Varley et al. (2024), where systems must adapt quickly to new, rare, or unseen scenarios without extensive annotation.

However, contemporary visual-language models such as CLIP Radford et al. (2021), despite their success in linking textual and visual modalities, remain heavily dependent on large-scale datasets.

Their performance degrades in complex scenes and under zero-shot conditions, revealing two key challenges. First, these models often lack robust reasoning capabilities and focus too narrowly on the most discriminative cues, leading to overfitting. Second, prior methods frequently address complexity by scaling dataset size Li et al. (2022b); Cheng et al. (2024); Liu et al. (2023), rather than improving the model's inherent ability to generalize from limited examples.

Efforts to enhance generalization through masking strategies have emerged. For instance, Masked Autoencoders He et al. (2022) and related approaches improve performance by reconstructing masked inputs, but they rely on random masking patterns that offer limited interpretability and control. More recent work Yang et al. (2023); Wei et al. (2024); Liang & Larson (2024) attempts to exclude primarily background regions, thereby reducing computational overhead and shifting attention onto the main object. Yet, simply removing irrelevant areas and revealing the entire salient region does not fundamentally improve a model's capacity to reason about partially observed features, hindering its true generalization potential.

To address these challenges, we propose **IMAGE**, a novel adaptive masking strategy designed to enhance model generalization in low-shot visual grounding tasks. Inspired by human perception, where objects are often identified from partial glimpses of distinctive features (e.g., the nose or wing of an airplane), IMAGE adaptively masks salient visual regions instead of backgrounds, as done in state-of-the-art methods Yang et al. (2023); Liang & Larson (2024); Wei et al. (2024). This strategy compels the model to infer objects from incomplete visual cues, encouraging deeper cognitive reasoning and facilitating robust, transferable representation learning, thereby improving generalization capabilities.

We thoroughly validate our method on datasets such as COCO Lin et al. (2014) and ODinW, demonstrating consistent improvements in zero-shot and low-shot scenarios without increasing computational overhead. Beyond enhancing performance, IMAGE offers a theoretically sound and empirically validated mechanism for training models to generalize under low-shot conditions, mitigating the need for ever-expanding datasets. Our contributions are as follows:

- We propose **IMAGE**, a novel adaptive masking paradigm that masks partial key features, forcing models to reason over residual features, and theoretically and empirically demonstrate its effectiveness on visual grounding tasks.

- We propose **RFGAM**, an effective masking modeling method that enables smooth, spatially coherent transitions between hard and soft masks, Offering a novel and effective way for mask modeling.

- We provide empirical evidence on standard benchmarks showing that IMAGE outperforms baseline and state-of-the-art strategies in low-shot settings, enhancing low-shot performance without computational overhead.

.

## 2 RELATED WORK

**Zero-Shot and low-shot Learning in Visual Grounding**   Zero-shot learning (ZSL) aims to recognize unseen classes by transferring knowledge from seen classes, enhancing generalization to novel categories Lampert et al. (2009); Farhadi et al. (2009); Socher et al. (2013). Early ZSL approaches employed attribute-based techniques and semantic embeddings to bridge the gap between seen and unseen classes Akata et al. (2015); Xian et al. (2018). With large-scale vision-language models such as CLIP Radford et al. (2021), recent studies have explored zero-shot grounding by leveraging pretrained architectures Gu et al. (2021); Li et al. (2022b). However, these approaches often rely on extensive datasets for pre-training and fine-tuning, limiting scalability. In contrast, low-shot learning focuses on learning from a small number of labeled examples Fei-Fei et al. (2006); Snell et al. (2017), and dedicated methods in visual grounding Kang et al. (2019); Sun et al. (2021) enhance generalization with limited annotations. Despite these advances, overfitting remains a challenge due to data scarcity, and many low-shot strategies utilize complex meta-learning frameworks Finn et al. (2017); Li et al. (2019).

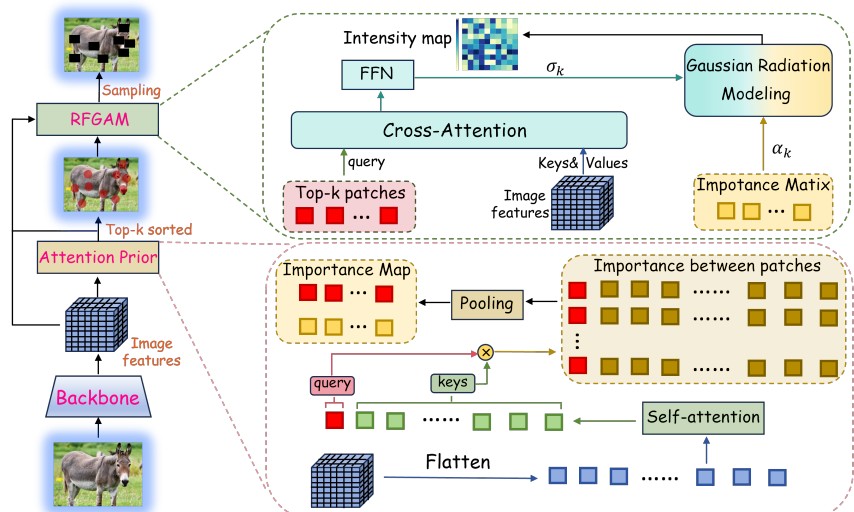

Figure 2: Pipeline of IMAGE method, consisting of two blocks: attention prior generation module and RFGAM mask generation module.

**Attention Mechanisms and Masking Strategies in Vision Models**    Attention mechanisms selectively focus on relevant input regions, improving interpretability and feature representation Bahdanau et al. (2015); Vaswani et al. (2017). In vision transformers, self-attention facilitates long-range dependencies, enhancing feature learning Dosovitskiy et al. (2021); Liu et al. (2021). Meanwhile, masking techniques in self-supervised learning (e.g., Masked Autoencoders He et al. (2022) and BEiT Bao et al. (2021)) have proven effective for representation learning by masking random patches or tokens. However, such methods generally employ random masking without specific guidance to emphasize crucial object features.

**Radiance Field Modeling and Gaussian Approaches**    Radiance fields are widely used to model high-fidelity scenes by parameterizing volumetric light interactions Mildenhall et al. (2020); Niemeyer et al. (2020). Neural Radiance Fields (NeRF)Mildenhall et al. (2020) leverage neural networks to represent continuous volumetric scenes for realistic rendering. Gaussian-based radiance modeling offers smoother representationsWang et al. (2021); Kim et al. (2022). Zhou et al. Zhou et al. (2016) demonstrated that global average pooling can localize discriminative regions without explicit localization supervision. Building on these insights, we adopt a dynamic Gaussian-based approach to model an importance prior distribution over feature maps. This flexibility enables adaptive masking of salient regions, avoiding rigid thresholds and enhancing the model's focus on key features.

## 3    METHODS

IMAGE enhances zero-shot and low-shot visual grounding without relying on scaling up dataset size. Unlike conventional masking approaches, IMAGE strategically obscures salient regions, compelling models to reason effectively from partial observations, as is shown in Fig. 2. Specifically, IMAGE introduces two core modules: an **Importance Prior Generation Block** ($\theta_p$), which estimates patch importance via global contextual interactions, and an **Adaptive Mask Generation Block** ($\theta_m$), which generates adaptive masks guided by the importance prior. Applied on hierarchical feature maps extracted from an image encoder, IMAGE dynamically directs model attention towards key semantic cues, thereby improving generalization performance without increasing data requirements.

### 3.1    IMPORTANCE PRIOR GENERATION

To guide adaptive masking, we generate an importance prior highlighting the salient regions in the image feature maps. Given a hierarchical feature map $F_i \in \mathbb{R}^{B \times H_i \times W_i \times C_i}$, representing batch size

$B$, number of channels $C_i$, and spatial dimensions $H_i \times W_i$ at scale $i$, we reshape it into spatial tokens $T_i \in \mathbb{R}^{B \times N_i \times C_i}$, with $N_i = H_i \times W_i$, each representing a distinct spatial patch.

With a self-attention mechanism, we encode global contextual relationships among these tokens, resulting in enriched embeddings. Each token's global relevance is quantified by aggregating its correlation with the global context into a scalar importance score. We normalize these scores to obtain a coherent importance prior, denoted as $\widetilde{S}_{\text{whole}} \in [0, 1]^{B \times N_i}$, effectively guiding the subsequent adaptive masking process.

## 3.2 Adaptive Mask Generation

Guided by the importance prior $\widetilde{S}_{\text{whole}}$, our adaptive mask generation strategy selectively obscures salient regions within images, enhancing model robustness to partial observations. Specifically, we combine a threshold-based masking scheme, which explicitly masks patches based on saliency ranking, with Radiance Field Gaussian Adaptive Masking (RFGAM), a probabilistic approach employing Gaussian radiance fields for smooth, coherent mask transitions.

**Threshold-Based Adaptive Masking**  We first rank patches at each feature-map scale $i$ according to their normalized importance $\widetilde{S}_{\text{whole}}$. The top $\rho_i\%$ are deemed critical, and we randomly mask a fraction ($\gamma\%$) of these critical patches. For the remaining patches, we apply random masking to reach the overall ratio $\rho_i\%$. This selective yet partially randomized scheme pushes the model to rely on truly essential patches, while ensuring it still has enough visible context to stabilize learning.

**Radiance Field Gaussian Adaptive Masking (RFGAM)**  Although the threshold-based strategy effectively isolate salient regions, their discrete boundaries can underuse the spatial continuity inherent in semantic structure. To mitigate this issue, we introduce RFGAM, which models the spatial importance landscape as a continuous radiance field parameterized by learned Gaussian kernels. Concretely, we combine the top-k patches features $f_k$ with their cross-attention outputs $c_k$ to learn the variance $\sigma_k^2$ of each kernel, which can be denoted as $\sigma_k^2 = \text{ReLU}(\text{FFN}_\sigma([f_k, c_k])) + \epsilon$ where $\epsilon$ as a regularization term. Given these variances and their corresponding importance weights $\alpha_k^{(b)}$, we construct a continuous intensity field across spatial coordinates $(x, y)$:

$$I^{(b)}(x, y) = \sum_{k=1}^{K_i} \alpha_k^{(b)} \exp\left(-\frac{\|(x, y) - (x_k, y_k)\|^2}{2\sigma_k^{2(b)}}\right).$$

Let $\mu(b)$ and $\sigma(b)$ denote the mean and standard deviation of $I^{(b)}(x, y)$. We define two thresholds:

$$T_{\text{hard}}^{(b)} = \mu(b) + [\delta + k]\sigma(b), \quad T_{\text{vis}}^{(b)} = \mu(b) - [\delta - k]\sigma(b),$$

where $\delta$ is a hyperparameter, and $k$ is a progressive learning offset that decays over epochs. With two thresholds, we can partition regions into hard masked, soft masked, or fully visible states. By combining flexible masking strategies with a progressive learning scheme, we enable the vision model to reason and learn robust features when salient features are partially missing while avoiding excessive masking that could hinder effective training. More implementation details are provided in Appendix.

**Optimization and Learning Strategy**  We do not introduce additional losses tied directly to masking. Instead, we follow Grounding DINO Liu et al. (2023), relying solely on contrastive loss for vision-language alignment and a localization loss for bounding box predictions. By tuning hyperparameters such as initial masking ratios and RFGAM thresholds, we balance the trade-off between visible information and the challenge of inference under occlusion. This strategy enhances generalization without enlarging the dataset, ultimately yielding stronger representations from limited data.

## 3.3 Theoretical Analysis

**Setup:**  Let $\mathcal{X}$ be the input space and $\mathcal{Y}$ be the output space. Assume there is a subset of highly discriminative features (or regions) indexed by $I \subseteq \{1, \ldots, d\}$. Based on how we handle $I$, we define three hypothesis classes:

1. $\mathcal{H}_{\text{dep}}$: No masking is applied. The model is free to exploit all features, including $I$, potentially leading to overfitting.

2. $\mathcal{H}_{\text{rob}}^{\text{(R-I)}}$: An "$R$-$I$" masking scheme, where all features, including those in $I$, may be masked according to some distribution $\mu_M^{\text{(R-I)}}$. Any $h$ in this class must maintain low loss under this random masking:

$$\forall (x, y): \quad \mathbb{E}_{M \sim \mu_M^{\text{(R-I)}}} \Big[ \ell\big(h(x \odot M), y\big) \Big] \ \leq \ \varepsilon.$$

3. $\mathcal{H}_{\text{rob}}^{\text{(A-I)}}$: An "$A$-$I$" masking scheme, where features in $I$ are masked with high probability (or exclusively), making it much harder for the model to rely solely on $I$. Formally, there is a distribution $\mu_M^{\text{(A-I)}}$ that masks or perturbs features in $I$ frequently:

$$\forall (x, y): \quad \mathbb{E}_{M \sim \mu_M^{\text{(A-I)}}} \Big[ \ell\big(h(x \odot M), y\big) \Big] \ \leq \ \varepsilon.$$

By definition, we have:

$$\mathcal{H}_{\text{rob}}^{\text{(A-I)}} \ \subseteq \ \mathcal{H}_{\text{rob}}^{\text{(R-I)}} \ \subseteq \ \mathcal{H}_{\text{dep}}.$$

**Rademacher Complexity:**  For a hypothesis class $\mathcal{H}$ and a sample $S = \{(x_i, y_i)\}_{i=1}^N$, the empirical Rademacher complexity is

$$\hat{R}_S(\mathcal{H}) \ = \ \mathbb{E}_\sigma \Big[ \sup_{h \in \mathcal{H}} \frac{1}{N} \sum_{i=1}^N \sigma_i \, h(x_i) \Big],$$

where $\sigma_i$ are i.i.d. Rademacher random variables taking values in $\{+1, -1\}$ with equal probability. The expected Rademacher complexity $R_N(\mathcal{H})$ is the expectation of $\hat{R}_S(\mathcal{H})$ over the random choice of $S$. A key property is monotonicity: if $\mathcal{H}_1 \subseteq \mathcal{H}_2$, then

$$R_N(\mathcal{H}_1) \ \leq \ R_N(\mathcal{H}_2).$$

**Complexity Reduction:**

**Lemma 1** (Complexity Reduction). *Since $\mathcal{H}_{rob}^{\text{(A-I)}} \subseteq \mathcal{H}_{rob}^{\text{(R-I)}} \subseteq \mathcal{H}_{dep}$, it follows that $R_N\big(\mathcal{H}_{rob}^{\text{(A-I)}}\big) \leq R_N\big(\mathcal{H}_{rob}^{\text{(R-I)}}\big) \leq R_N\big(\mathcal{H}_{dep}\big)$.*

*Proof of Lemma 1.*  The containment relations imply these Rademacher complexity inequalities via monotonicity. Intuitively, $\mathcal{H}_{\text{dep}}$ may include overfitted solutions that exploit $I$ aggressively, $\mathcal{H}_{\text{rob}}^{\text{(R-I)}}$ excludes some of these by masking features at random, and $\mathcal{H}_{\text{rob}}^{\text{(A-I)}}$ imposes the strictest requirement by frequently masking the most discriminative set $I$. Hence, the complexity is successively reduced in these classes. $\qquad\square$

**Generalization Bounds:**  A standard Rademacher-based generalization result states: given any $\delta > 0$, with probability at least $1 - \delta$,

$$L(h) \ \leq \ \hat{L}_S(h) \ + \ 2\,R_N(\mathcal{H}) \ + \ O\Big(\sqrt{\tfrac{\log(1/\delta)}{N}}\Big),$$

for all $h \in \mathcal{H}$. Here, $L(h)$ is the expected loss and $\hat{L}_S(h)$ is the empirical loss.

**Theorem 1** (Better Generalization from Adaptive Masking). *Let $h_{dep} \in \mathcal{H}_{dep}$, $h_{rand} \in \mathcal{H}_{rob}^{\text{(R-I)}}$, $h_{targ} \in \mathcal{H}_{rob}^{\text{(A-I)}}$ achieve similar empirical performance $\hat{L}_S(h_{dep}) \approx \hat{L}_S(h_{rand}) \approx \hat{L}_S(h_{targ})$ on the training sample $S$. Suppose that $R_N\big(\mathcal{H}_{rob}^{\text{(A-I)}}\big) < R_N\big(\mathcal{H}_{rob}^{\text{(R-I)}}\big) < R_N\big(\mathcal{H}_{dep}\big)$. Then the generalization bound for $h_{targ}$ is strictly tighter than those for $h_{rand}$ and $h_{dep}$. Consequently, $L(h_{targ}) < L(h_{rand}) < L(h_{dep})$, showing that adaptive masking of key features improves generalization more than random masking or no masking.*

Table 1: Comprehensive comparison of IMAGE variants with baseline and SOTA across different low-shot settings.

| Method | COCO val2017 (Close-set) | ODinW_13 (Zero-shot) | ODinW_35 (Zero-shot) | COCO val2017 (low-shot) |
|---|---|---|---|---|
| Baseline | 0.454 | 0.208 | 0.092 | 0.400 |
| RandomMask | 0.456 | 0.190 | 0.085 | 0.392 |
| MaskCLIP Yang et al. (2023) | 0.465 | 0.212 | 0.089 | 0.401 |
| CenterMask Liang & Larson (2024) | 0.473 | 0.225 | 0.094 | 0.405 |
| ClusterMask Wei et al. (2024) | 0.461 | 0.206 | 0.089 | 0.408 |
| IMAGE | 0.473 | 0.235 | 0.104 | 0.426 |
| **IMAGE(RG)** | **0.481 (+2.7%)** | **0.251 (+4.3%)** | **0.112 (+2.0%)** | **0.437 (+3.7%)** |

*Proof of Theorem 1.* From the standard Rademacher bound, we have

$$L(h_{\text{dep}}) - \hat{L}_S(h_{\text{dep}}) \ \leq \ 2\,R_N\big(\mathcal{H}_{\text{dep}}\big) \ + \ O\big(\sqrt{\log(1/\delta)/N}\big),$$

$$L(h_{\text{rand}}) - \hat{L}_S(h_{\text{rand}}) \ \leq \ 2\,R_N\big(\mathcal{H}_{\text{rob}}^{(\text{R-I})}\big) \ + \ O\big(\sqrt{\log(1/\delta)/N}\big),$$

$$L(h_{\text{targ}}) - \hat{L}_S(h_{\text{targ}}) \ \leq \ 2\,R_N\big(\mathcal{H}_{\text{rob}}^{(\text{A-I})}\big) \ + \ O\big(\sqrt{\log(1/\delta)/N}\big).$$

Given similar empirical losses, the lower complexity term for $\mathcal{H}_{\text{rob}}^{(\text{A-I})}$ yields a tighter bound on $L(h_{\text{targ}})$, followed by $\mathcal{H}_{\text{rob}}^{(\text{R-I})}$, and finally $\mathcal{H}_{\text{dep}}$. $\qquad\qquad\square$

**Remark 1.** *This result shows that forcing a model to train without always relying on the most discriminative features (i.e., via high-probability masking of I) yields a strictly smaller hypothesis space and thus better theoretical guarantees. It also indicates that merely random masking of all features (including those in I) is beneficial but not as effective as targeting I directly. Empirically, this aligns with improved robustness and generalization when critical features are deliberately masked during training.*

## 4 EXPERIMENTS

### 4.1 EXPERIMENTAL SETUPS

**Dataset** Experiments are conducted on COCO and ODinW datasets. COCO (*train2017/val2017*) is utilized for closed-set training and evaluation. For zero-shot detection, ODinW datasets (ODinW_13 and ODinW_35) containing unseen object categories are employed. low-shot experiments are performed by randomly sampling subsets comprising 5%, 10%, 20%, and 30% of COCO *train2017* data.

**Evaluation Metrics** Standard COCO Average Precision (AP) is adopted for closed-set evaluations. For zero-shot detection on ODinW datasets, we report Average Precision (AP) aggregated across all unseen categories without explicit per-class averaging. low-shot performance is evaluated using AP on COCO *val2017*.

**Implementation Details** Our method builds upon Grounding Dino with a Swin-T backbone, integrating adaptive masking modules after each feature extraction stage. Initial masking rates for the four backbone layers are set at 20%, 30%, 40%, and 50%, respectively. In the RFGAM module, the Gaussian modeling parameter $k_0$ decays gradually from 0.5 to near-zero throughout training. The learning rate is set to 0.005, and training is performed on 8 RTX4090 GPUs. By default, IMAGE uses masks sampled directly from learned importance priors. IMAGE(RG) applies an additional Gaussian radiance modeling step to smooth the spatial distribution of importance scores.

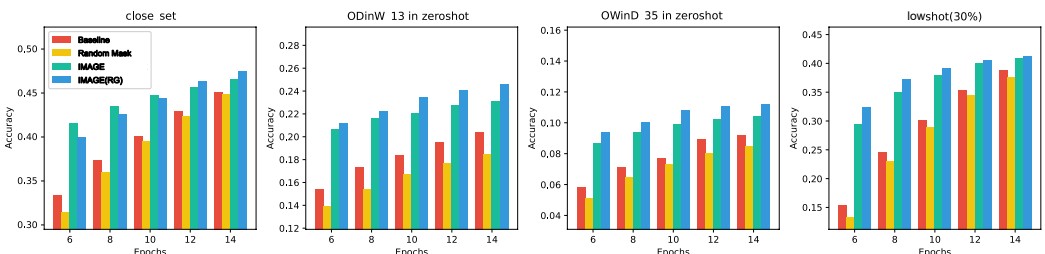

Figure 3: Scaling laws of our IMAGE method. With increased epochs, IMAGE achieves more accurate grounding AP across all four datasets and three settings.

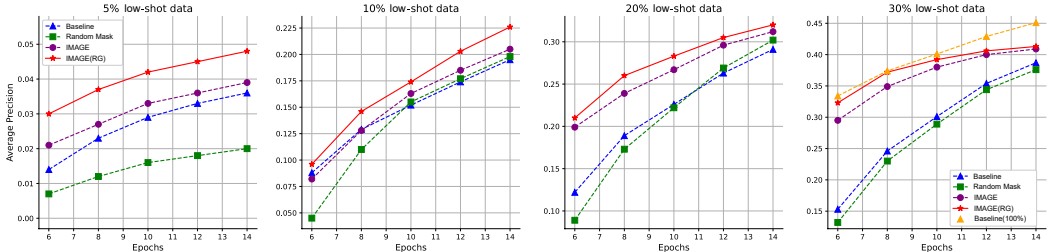

Figure 4: Comparison between IMAGE with other strategies in different low-shot ratios.

## 4.2 QUANTITATIVE RESULTS

**Overall Performance**  We comprehensively evaluated IMAGE under zero-shot, close-set, and low-shot visual grounding settings. As is illustrated in Table 1, an ablation study on COCO *val2017* clearly demonstrated that our methods greatly enhances generalization. IMAGE(RG) achieved 0.481 AP, outperforming the Baseline (no masking) by a notable 2.7%, RandomMask by 2.5%, and the simpler threshold-based IMAGE by 1.4%. Notably, in challenging zero-shot conditions, IMAGE(RG) exhibited remarkable improvements, with gains of 4.3% and 2.0% AP on ODinW_13 and ODinW_35 datasets, respectively. Furthermore, under low-shot conditions, IMAGE(RG) consistently maintained superior performance, surpassing the Baseline by 3.7% AP, validating its robust capability to infer critical visual cues with minimal labeled data.

To further demonstrate superiority, we compared IMAGE(RG) against state-of-the-art masking methods, including MaskCLIP Yang et al. (2023), CenterMask Liang & Larson (2024), and Cluster-Mask Wei et al. (2024). These methods primarily mask peripheral or background regions, whereas IMAGE(RG) uniquely and strategically obscures even portions of the most salient areas, compelling models to develop stronger contextual reasoning. As shown in Table 1, IMAGE(RG) consistently outperformed all these methods across all settings, achieving an AP of 0.251 on ODinW_13, surpassing MaskCLIP by 2.6%, CenterMask by 2.1%, and ClusterMask by 1.8%. Moreover, in the low-shot scenario on COCO *val2017*, IMAGE(RG) exceeded these leading methods by at least 2.9% AP, demonstrating clear advantages in robustness, interpretability, and computational efficiency.

**Scaling Laws across All Settings**  To further illustrate the effectiveness of IMAGE(RG), we analyzed its scaling behavior throughout training, as shown in Figure 3. Notably, IMAGE(RG) achieves faster convergence and higher grounding AP compared to Baseline and RandomMask methods across all datasets and learning scenarios. Even at early training stages (e.g., epochs 6–10), IMAGE(RG) consistently demonstrates a substantial performance gap, improving baseline methods by over 18% AP in low-shot scenarios and similarly significant margins in close-set and zero-shot scenarios. This accelerated early-stage performance reflects its unique ability to prioritize critical semantic regions effectively from the outset, ensuring a robust foundation for representation learning.

Moreover, IMAGE(RG) continues to outperform other methods as training progresses, maintaining its advantage over both basic masking strategies and the simpler IMAGE variant. Such sustained improvement highlights the long-term benefits of strategically directing attention to essential features

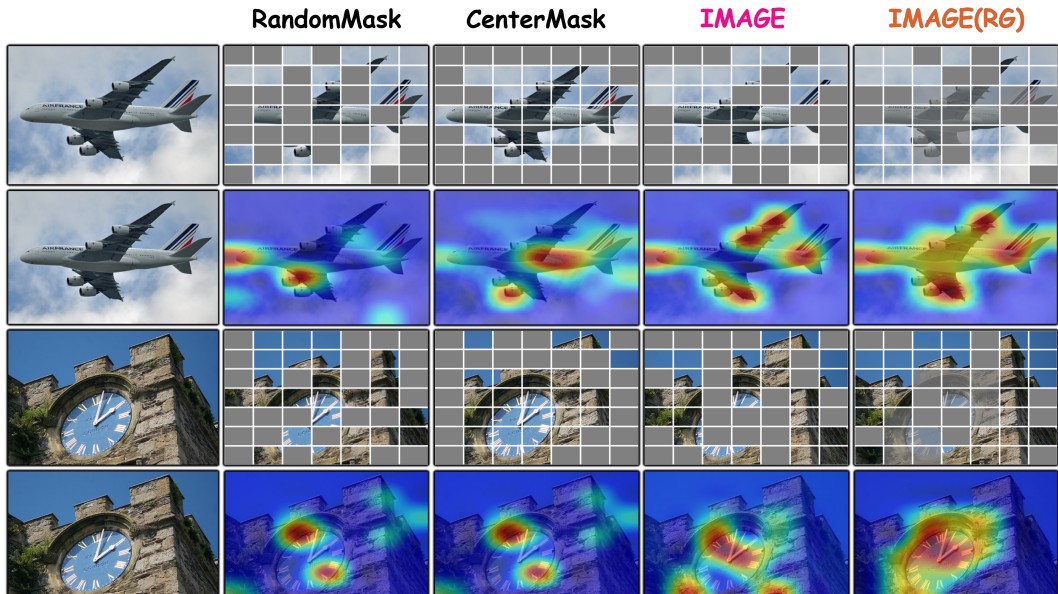

Figure 5: Visualization of both generated masks and importance(upper row) distributions (lower row) for images under different masking methods: RandomMask, CenterMask, IMAGE, and IMAGE(RG). Compared to other methods, IMAGE(RG) yields more concentrated and adaptive coverage, effectively highlighting the entire salient region necessary for accurate grounding, while the RandomMask and CenterMask produce relatively dispersed attention.

| Masking Ratio (%) | COCO val2017 | ODinW_13 | ODinW_35 | COCO val2017 (low-shot) |
|---|---|---|---|---|
| 10–40 | 0.479 | 0.248 | 0.109 | 0.431 |
| 20–50 | **0.481** | 0.251 | **0.112** | **0.437** |
| 30–60 | 0.470 | **0.253** | 0.111 | 0.424 |

Table 2: IMAGE results across different datasets and masking ratio ranges.

early during training, confirming that the adaptive Gaussian-based masking strategy is effective not only for rapid convergence but also for achieving stable and superior representations over extended training cycles.

**low-shot Training with Different Data Ratios**  To assess IMAGE's robustness under limited annotated data, we conducted low-shot experiments on COCO with varying fractions (5%, 10%, 20%, and 30%) of the *train2017* set. As shown in Fig. 4, IMAGE(RG) consistently achieves significant performance improvements over baseline methods, especially in scenarios with severe data scarcity. For instance, using only 30% training data after merely 6 epochs, IMAGE(RG) attains an AP of **32.3%**, substantially exceeding the baseline's **15.3%**. This remarkable performance gain highlights IMAGE(RG)'s capability to quickly learn robust representations by adaptively prioritizing essential visual cues, thereby enhancing both generalization and convergence efficiency under practical, resource-constrained conditions.

**Impact of Different Mask Ratios**  We evaluate how varying initial mask ratios affect performance, testing three schemes (10–40%, 20–50%, 30–60%) across four encoder layers. As shown in Table 2, the 20–50% range consistently yields the best AP scores, suggesting that moderately balanced masking improves feature quality. While optimal ratios may vary slightly by dataset, 20–50% offers robust generalization.

## 4.3 ABLATION STUDIES

**Impact of Importance Prior in Adaptive Masking**   We analyze the role of our importance prior by comparing against random masking and a no-masking baseline. As shown in Table 1, importance-guided masking consistently outperforms the alternatives across close-set, zero-shot, and low-shot scenarios. In particular, its gains in zero-shot settings highlight its ability to generalize to unseen categories by focusing on semantically critical regions and encouraging robust inference from partial yet salient cues.

**Benefit of Gaussian Radiance Field Modeling**   To assess the effect of Gaussian-based radiance modeling, we compare IMAGE(RG) with standard IMAGE. Table 1 shows that RFGAM yields more nuanced and spatially coherent masks, leading to consistent improvements in zero-shot and low-shot performance. These results demonstrate that smooth spatial transitions enhance both feature expressiveness and robustness in limited-data scenarios.

| Settings | Close-set | low-shot | Zero-shot |
|:---:|:---:|:---:|:---:|
| non-progressive | 0.476 | 0.426 | 0.235
0.110 |
| progressive | **0.481** | **0.437** | **0.251**
**0.112** |

Table 3: The comparison of non-progressive and progressive training across settings. For Zero-shot, results are reported for ODinW_13 (upper) and ODinW_35 (lower).

**Effectiveness of Progressive Training Strategy**   We assess the impact of progressively decaying the parameter $k$ in IMAGE(RG) by comparing it with a fixed-$k$ baseline. As shown in Table 3, progressive decay consistently improves AP across COCO *val2017* (+0.5%), ODinW_13 zero-shot (+1.6%), and 30% low-shot (+1.1%). These gains suggest that gradual reduction of $k$ encourages adaptive masking, leading to better feature learning and generalization.

## 4.4 QUALITATIVE ANALYSIS

Figure 5 visualizes masking patterns and importance scores for two examples (airplane and clock). RandomMask yields scattered attention with little semantic focus, while CenterMask roughly outlines object regions but misses key discriminative parts (e.g., plane's nose or clock dial), reflecting limited selectivity and generalization.

In contrast, IMAGE masks moderately salient regions to force reasoning from core cues, leading to more focused representations. IMAGE(RG) further improves by blending soft and hard masking via RFGAM, producing smoother attention and more adaptive inference. These results suggest that IMAGE and IMAGE(RG) enable more targeted, flexible masking, enhancing robustness and generalization in visual grounding.

## 5 CONCLUSION

In this paper, we proposed IMAGE, a novel method for improving zero-shot and low-shot visual grounding without increasing dataset sizes. Inspired by cognitive science and Masked Autoencoders He et al. (2022), IMAGE adaptively masks salient regions in feature maps generated by the vision backbone, compelling the model to reason from remaining critical information. This approach results in robust and generalized representations capturing both local and global features. Experimental evaluations on benchmark datasets, including COCO and ODinW, demonstrate IMAGE's superior performance compared to existing methods in both zero-shot and low-shot scenarios. Our findings underscore adaptive masking, leveraging attention mechanisms and Gaussian modeling methods, as an effective alternative to typical data augmentation strategies. This work advances low-shot learning and opens avenues for future research into efficient and interpretable visual grounding.

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

# A APPENDIX

## A.1 IMPLEMENTATION DETAILS

**Progressive Masking Strategy**  As training advances, we increase masking difficulty step-by-step. Lower-level feature maps retain more visible detail initially, aiding early representation learning, while higher-level maps receive heavier masking to encourage the model to reason abstractly. Simultaneously, we anneal $k$ over epochs:

$$k_{\text{epoch}} = k_0 \left( 1 - \frac{\text{epoch}}{E_{\text{total}}} \right),$$

intensifying masking as the model matures. This progressive approach allows the network to gradually adapt to greater occlusion, improving its ability to infer missing content and develop robust, context-driven representations.

**Soft Mask Scheme**  Given $I^{(b)}(x,y)$ relative to $T_{\text{hard}}^{(b)}$ and $T_{\text{no-mask}}^{(b)}$, each patch is assigned a mask value:

$$M_i^{(b,p)} = \begin{cases} 0, & \text{if } I^{(b)}(x,y) > T_{\text{hard}}^{(b)}, \\ 1, & \text{if } I^{(b)}(x,y) < T_{\text{no-mask}}^{(b)}, \\ 1 - \frac{I^{(b)}(x,y) - T_{\text{no-mask}}^{(b)}}{T_{\text{hard}}^{(b)} - T_{\text{no-mask}}^{(b)}}, & \text{otherwise.} \end{cases}$$

This formulation smoothly interpolates between fully masked and fully visible states, respecting the continuous nature of the underlying importance field. Applying $M_i$ to the feature map $F_i$ via element-wise multiplication ($F_i' = F_i \odot \text{Reshape}(M_i)$) ensures that highly informative regions receive less masking, while less critical areas are increasingly obscured.

**Training Time and Memory Overhead**  Compared with the baseline training pipeline, our IM-AGE(RG) method introduces only minimal extra computation. Training time increases by around 10%, which is mainly caused by the additional attention-based mask generation process. In terms of memory, consumption grows by about 100MB, largely due to maintaining the radiance prior and performing Gaussian-based mask computations.

## A.2 MORE EVALUATION RESULTS

**Experiments on Ultra-Low Data Regimes (1-2%)**  In addition to the main results (see Section 4.2 and Fig. 4 of the paper), we further evaluate data efficiency in an ultra-low data regime. Specifically, we train all methods using only 1% of the labeled data. As shown in Table 4, despite the instability typically observed at such extreme sparsity, our approach remains robust and clearly outperforms baselines.

Table 4: Performance at **1%** labeled data.

| Method | Baseline | Random Mask | IMAGE | IMAGE(RG) |
|--------|----------|-------------|-------|-----------|
| AP @ 1% | 1.90 | 0.95 | 2.60 | **3.30** |

These results indicate that both IMAGE and IMAGE(RG) maintain clear advantages even when supervision is extremely scarce. In particular, IMAGE(RG) achieves the best performance, suggesting that the paradigm is especially beneficial in the ultra-low data regime.

**Generalizability Across Model Paradigms**  To evaluate the generalizability of the proposed IMAGE framework across different model paradigms, we extend our analysis beyond DETR-style architectures. While the main body of experiments is conducted on Grounding DINO (Swin-T), a representative DETR-based visual grounding model, we further investigate the applicability of IMAGE on YOLO-based models.

Specifically, we integrate IMAGE into YOLO-World and perform evaluations on several benchmarks. As shown in Table 5, Across all evaluation settings, IMAGE yields consistent performance gains over the YOLO-World baseline. This demonstrates that the proposed approach is architecture-agnostic and effectively enhances visual grounding performance across heterogeneous backbones.

Table 5: Generalization of IMAGE to YOLO-based architecture (YOLO-World). Results are reported on COCO `val2017` (closed-set and low-shot), ODinW-13, and ODinW-35.

| Method | COCO val2017 | ODinW-13 | ODinW-35 | COCO val2017 (Low-shot) |
|---|---|---|---|---|
| YOLO-World | 0.412 | 0.177 | 0.090 | 0.373 |
| IMAGE | 0.448 | 0.221 | 0.101 | 0.415 |
| IMAGE (RG) | **0.463** | **0.234** | **0.112** | **0.427** |

**Comparison with Semantic Masking Strategies**  To further evaluate the effectiveness of the proposed masking paradigm, we conduct an additional comparison with a strong semantic-aware masking method, SemMAE Li et al. (2022a). This comparison complements existing evaluations against widely-used masking baselines such as random masking (MAE), CenterMask, and ClusterMask.

The evaluation follows the same training and testing protocols described in the main paper. As shown in Table 6, both IMAGE and its variant IMAGE (RG) achieve consistently higher performance than SemMAE across all benchmarks. These results highlight the effectiveness of our masking paradigm, which achieves superior performance without relying on external semantic priors or additional model complexity.

Table 6: Performance comparison with SemMAE across four benchmarks.

| Method | COCO (Close) | ODinW-13 | ODinW-35 | COCO (Low-shot) |
|---|---|---|---|---|
| SemMAE | 0.469 | 0.219 | 0.091 | 0.412 |
| IMAGE | 0.473 | 0.235 | 0.104 | 0.426 |
| IMAGE (RG) | **0.481** | **0.251** | **0.112** | **0.437** |

