# OpenReview forum: "Adaptive Masking Enhances Visual Grounding"
_ICLR.cc/2026/Conference — ICLR 2026 Conference Withdrawn Submission_

### Official Review · Reviewer_F5W1 · 2025-10-24

**Soundness:** 2
**Presentation:** 3
**Contribution:** 2
**Rating:** 2
**Confidence:** 5

**Summary:**

This paper proposes IMAGE (Interpretative MAsking with Gaussian Radiation ModEling), a novel training paradigm for visual grounding that enhances model robustness and generalization, particularly in low-shot and zero-shot scenarios. Inspired by human perception, which can recognize objects from partial cues, IMAGE strategically masks salient regions of an image during training, forcing the model to learn to infer objects from incomplete information. The method introduces a progressive training strategy and a Gaussian-based mask generation module (RFGAM) to create smooth, spatially coherent masks. Experiments show that IMAGE significantly improves performance on standard benchmarks (COCO, ODinW) without adding computational cost at inference time. This paper draws on the powerful conceptual essence of "robust learning through progressive data corruption" from diffusion models, but creatively applies and adapts it.

**Strengths:**

1. The paper is well-motivated. Masking salient regions instead of backgrounds is intuitive.

2. The paper provide theoretical foundation making it soundness.

**Weaknesses:**

1. The paper only focus on referring expression comprehension
(REC) in visual grounding (VG), without attention about referring expression generation (REG) in VG.

2. Too many related works and compared methods in visual grounding are missed, such as [1-6]. It is suggested to add more detailed discussions and comparsions.

3. Besides, more comprehensive benchmarks should be added, including RefCOCO/+/g, Visual Grenome and FineCops-Ref.

4. Meanwhile, most powerful VLMs (e.g., Qwen-VL, InternVL, GPT-4V) that are not based on the Grounding DINO architecture. This makes it hard to gauge its impact in the broader landscape of VLM research.

5. The qualitative analysis in Figure 5 is brief. A deeper, more systematic analysis of *why* the model fails on certain examples, especially when masking is applied, would strengthen the paper.

[1] Yang, Xiaoyu, et al. "Enhancing visual grounding and generalization: A multi-task cycle training approach for vision-language models." arXiv preprint arXiv:2311.12327 2023.

[2] N. Wang, J. Deng, and M. Jia, “Cycle-consistency learning for captioning and grounding,” in Proceedings of the AAAI Conference on Artificial Intelligence, vol. 38, no. 6, 2024, pp. 5535–5543

[3] H. You, H. Zhang, Z. Gan, X. Du, B. Zhang, Z. Wang, L. Cao, S.-F. Chang, and Y. Yang, “Ferret: Refer and ground anything anywhere at any granularity,” in The Twelfth International Conference on Learning Representations, 2024.

[4] A. Zhang, Y. Yao, W. Ji, Z. Liu, and T.-S. Chua, “NExt-chat: An LMM for chat, detection and segmentation,” in Forty-first International Conference on Machine Learning, 2024.

[5] W. Wang, Q. Lv, W. Yu, W. Hong, J. Qi, Y. Wang, J. Ji, Z. Yang, L. Zhao, S. XiXuan, J. Xu, K. Chen, B. Xu, J. Li, Y. Dong, M. Ding, and J. Tang, “CogVLM: Visual expert for pretrained language models,” in The Thirty-eighth Annual Conference on Neural Information Processing Systems, 2024.

[6] L. Yang, X. Li, Y. Wang, X. Wang, and J. Yang, “Fine-Grained Visual
Text Prompting,” IEEE Transactions on Pattern Analysis and Machine
Intelligence, vol. 47, no. 3, pp. 1594–1609, 2025

**Questions:**

See weakness

---

### Official Review · Reviewer_5FTx · 2025-10-30

**Soundness:** 3
**Presentation:** 3
**Contribution:** 3
**Rating:** 6
**Confidence:** 5

**Summary:**

This paper proposes using saliency maps to guide the training of the Grounding DINO model, thereby enhancing its generalization ability. The authors use feature activation maps as priors and then adaptively mask the model to improve its generalization capabilities. The results are validated on object detection tasks, including closed-set, low-shot, and zero-shot capabilities, demonstrating the improvements of the proposed method.

**Strengths:**

- Using saliency maps to guide model training is interesting and important because it makes sense how to use interpretable results to improve model performance or fix model errors.
- The authors validated the effectiveness of the proposed method on several object detection tasks.

**Weaknesses:**

- Priors based on feature maps are often unreliable, especially in the ViT architecture. Although the attention map based on DINO v1 is more in line with human cognition, attention maps like CLIP or DINO v2 are not in line with human cognition, but still have good performance [1].
- The experiments in this paper appear to primarily focus on object detection tasks, utilizing only the visual grounding paradigm of Grounding DINO. I suggest the authors supplement this with experiments on referring expression comprehension (REC), using RefCOCO or other datasets.
- There is a formatting issue with the citation of references. Please note the difference between `\cite{}` and `\citep{}`.
- Some methods for interpretable guided training of object detection tasks are suggested to discuss in this paper [2,3]. Chen et al. [2] used counterfactual saliency maps for interpretable erasure to enhance generalization in the few-shot detection task. The authors should discuss the differences between this method and the one presented in this paper in the introduction.
- It is recommended to add some object detection interpretation references related to mask-based [4,5,6].
- The importance of the saliency map in Figure 5 is only qualitatively represented, lacking indicators such as insertion or deletion (if you want to prove that the importance found by the method in this paper is superior) [6].
- Typo (Line 85): "... and soft masks, **O**ffering a novel and ..." -> "... and soft masks, **o**ffering a novel and ..."
- Typo (Line 91): An extra period was added to the line following "contribution".
- Typo (Line 416): "low-shot" -> "Low-shot"



[1] Vision Transformers Need Registers. ICLR 2024.

[2] Bbam: Bounding box attribution map for weakly supervised semantic and instance segmentation. CVPR 2021.

[3] Generalized Semantic Contrastive Learning via Embedding Side Information for Few-Shot Object Detection. TPAMI 2025.

[4] Black-box Explanation of Object Detectors via Saliency Maps. CVPR 2021.

[5] Making Sense of Dependence: Efficient Black-box Explanations Using Dependence Measure. NIPS 2022.

[6] Interpreting Object-level Foundation Models via Visual Precision Search. CVPR 2025.

**Questions:**

- Some experimental details are missing. Is the zero-shot capability of the OdinW dataset the generalization capability of the method in this paper after training on COCO?

Suggestion:

- For future work, I suggest that the authors consider using a faithful attribution map [1] (instead of an activation graph or attention) to analyze the region that the model is currently focusing on, and then design feedback or enhancement mechanisms to improve the model's generalization ability.

[1] Interpreting Object-level Foundation Models via Visual Precision Search. CVPR 2025.

---

### Official Review · Reviewer_otVB · 2025-10-30

**Soundness:** 1
**Presentation:** 2
**Contribution:** 2
**Rating:** 2
**Confidence:** 3

**Summary:**

The paper proposes IMAGE, an adaptive masking paradigm designed to improve zero-shot and low-shot visual grounding. Instead of masking random or background regions, IMAGE selectively obscures salient visual regions, compelling models to infer from incomplete but informative cues - thus enhancing generalization and robustness. The method introduces a Radiance Field Gaussian Adaptive Masking (RFGAM) mechanism to create smooth, spatially coherent mask transitions and a progressive training strategy that increases masking difficulty over time. Experiments on COCO and ODinW benchmarks show that IMAGE and its variant IMAGE (RG) bring the performance improvement.

**Strengths:**

1. The paper provides some theoretical derivations and offers theoretical verification.

2. Experiments demonstrate the superiority of the proposed method over random masking.

**Weaknesses:**

1. The Introduction contains logical leaps, especially lacking clear explanations of key concepts and analysis of related works.

(1) The core of low-shot learning lies in the scarcity of samples - how is this related to feature missing?

(2) In low-shot/few-shot scenarios, why does the paper choose the masking-based approach for CLIP training? What are the main challenges faced by existing low-shot/few-shot CLIP methods?

(3) The core idea of this work is the adaptive masking of salient visual regions - how is this connected to the few-shot setting? Theoretically, would the proposed method also improve performance if trained on the full dataset?

In addition, [1] is highly relevant to this work and should be compared both conceptually and experimentally.

[1] Pei, G., Chen, T., Wang, Y., Cai, X., Shu, X., Zhou, T., & Yao, Y. (2025). Seeing What Matters: Empowering CLIP with Patch Generation-to-Selection. In Proceedings of the Computer Vision and Pattern Recognition Conference (pp. 24862-24872).

2. The claim may be problematic.
As far as I know, ClusterMask does not emphasize that background regions should be masked. For example, in Figure 1 (a) of the ClusterMask paper, the salient object is masked after applying ClusterMask rather than the background.

3. For the experimental comparison, the authors should conduct a thorough and fair evaluation against existing methods under standard settings.

(1) The low-shot setting used in this paper (selecting 5%–30% of the data) does not appear to be a common experimental setup. In few-shot learning, it is more typical to use K samples per class.

(2) In the zero-shot setting, the comparisons with existing methods are insufficient. On the one hand, the authors did not compare with the zero-shot results of existing works  (e.g., ClusterMask) on the COCO dataset. On the other hand, several datasets commonly used in prior works - such as Flickr8k, Flickr30k, and Caltech101 - are not included in the experiments.

4. The paper lacks detailed overhead results. Although the appendix mentions a 10% increase in runtime and a 100 MB increase in memory usage, it would be helpful to provide more detailed results - for example, what are the baseline runtime and memory values, and what are the corresponding values for IMAGE?

Overall, this paper lacks explanations of key concepts, making it difficult for readers to understand the insights behind the technical choices. In addition, the work fails to provide a clear comparison with related studies, raising concerns about the effectiveness of the method - specifically, whether it truly addresses the core issues of existing approaches. Finally, the use of non-standard experimental settings further weakens the verification for the effectiveness. Therefore, my rating is Reject now.

**Questions:**

Please refer to Weaknesses.

---

### Official Review · Reviewer_32KX · 2025-11-04

**Soundness:** 3
**Presentation:** 3
**Contribution:** 2
**Rating:** 4
**Confidence:** 4

**Summary:**

This paper presents a training paradigm for visual grounding inspired by the human ability to recognize objects under incomplete information. The proposed method selectively masks salient regions and progressively increases the masking ratio during training, encouraging models to infer object attributes from limited cues rather than memorizing all features. This approach enhances robustness and data efficiency, improving zero-shot and low-shot performance across standard visual grounding benchmarks.

**Strengths:**

The authors propose IMAGE, a method that, unlike conventional masking approaches, strategically obscures salient regions to encourage models to reason more effectively from partial observations.

**Weaknesses:**

The author mentions that the training strategy for occluding salient targets is proposed by drawing on human perception. However, human perception also relies on the relationship between the background and foreground to identify objects. It remains unclear how this aspect was considered or modeled in the proposed approach.
In the Adaptive Mask Generation Block, if the occlusion ratio is excessively high, could it lead the model to overfit certain features, thereby negatively affecting overall performance?
In the experimental section, the authors compared their approach only with other masking methods. It is recommended to include additional baseline models to more comprehensively demonstrate the contribution of the proposed occlusion strategy to the visual grounding community.

**Questions:**

Please refer to the Weaknesses.

---

### Note · Authors · 2025-11-14

I have read and agree with the venue's withdrawal policy on behalf of myself and my co-authors.